Corrected: Publisher Correction

# Chip-scale atomic diffractive optical elements

Liron Stern[1,2], Douglas G. Bopp[1,2], Susan A. Schima[1], Vincent N. Maurice [1,2] & John E. Kitching[1]

The efficient light–matter interaction and discrete level structure of atomic vapors made possible numerous seminal scientific achievements including time-keeping, extreme non-linear interactions, and strong coupling to electric and magnetic fields in quantum sensors. As such, atomic systems can be regarded as a highly resourceful quantum material platform. Recently, the field of thin optical elements with miniscule features has been extensively studied demonstrating an unprecedented ability to control photonic degrees of freedom. Hybridization of atoms with such thin optical devices may offer a material system enhancing the functionality of traditional vapor cells. Here, we demonstrate chip-scale, quantum diffractive optical elements which map atomic states to the spatial distribution of diffracted light. Two foundational diffractive elements, lamellar gratings and Fresnel lenses, are hybridized with atomic vapors demonstrating exceptionally strong frequency-dependent, non-linear and magneto-optic behaviors. Providing the design tools for chip-scale atomic diffractive optical elements develops a path for compact thin quantum-optical elements.

[1] National Institute of Standards and Technology, Time & Frequency Division, 325 Broadway, Boulder, CO 80305, USA. [2] Department of Physics, University of Colorado, Boulder, CO 80309, USA. Correspondence and requests for materials should be addressed to L.S. (email: liron.stern@nist.gov)

Diffractive optical elements and subwavelength dielectric–metallic elements (metasurfaces) are important building blocks in science and technology[1–6]. Such thin surfaces offer extraordinary control of the different degrees of freedom of light, such as phase, polarization, and spectral and spatial distributions[7–9]. Moreover, devices can be tailored to be actuated mechanically[10], all-optically[11], and by polarization control[12]. Indeed, numerous optical elements have been implemented in this way, including holograms[13], lenses[8,14], and Dammann gratings[15]. Atomic vapors are fundamental resources offering highly spectrally dependent and nonlinear control of the phase and amplitude of light both in the quantum and classical regimes. Indeed, the strongly resonant system offers nonlinear atom–photon interactions down to the few photon regime[16], extreme magneto-optic Verdet coefficients[17], and ultrasensitive quantum sensors, such as magnetometers and gyrosopes[18–20]. Moreover, the discrete levels provide long-lived clock transitions[21–23] and the associated steep dispersions allow large reduction in the speed of light[24].

Early combinations of atomic systems and thin surfaces have been demonstrated in the cases of dielectric[25] and metallic metasurfaces[26], which effectively show that by combining such optical elements with standalone vapor cells or table-top vacuum chambers, hybridization of spatial modes, as well as polarization control can be achieved.

In this paper, we present a fully integrated chip-scale atomic diffractive optical element (ADOE). The ADOE consists of channels etched into silicon which are filled with gaseous alkali atoms forming an atomic–dielectric grating. The simplicity of the technique of anodic bonding glass to silicon[19,22,27–29] lends itself well to wafer-level fabrication, providing dozens of devices in a single batch. By controlling the atomic state, the efficiency of the different diffraction orders can be tailored, thus mapping the atomic populations to the spatial diffraction pattern. The different spectra of the hybridized phase-amplitude atomic grating are measured and explained by a simple model. The hybrid element gains the steep dispersive properties near of the atomic transitions while maintaining the rich designability of diffractive optical elements. Indeed, we further demonstrate an atomic Fresnel lens, capable of switching the efficiency of the focusing power by controlling the atomic state. Such a lens offers more than 95% contrast with potential atomic lifetime-limited switching speeds in the mid-MHz regime. The versatility of the parameter space to control our ADOEs is demonstrated by revealing the optical nonlinearity and magneto-optic response of our system. The fabrication process is highly scalable, and may be easily adapted to incorporate metasurfaces or thin optical element with rubidium atoms; moreover, due to the atomic confinement, all the atoms in the device contribute to the device behavior with eliminated background contribution. As such, the concept may pave the way to a variety of quantum controlled, chip-scale thin elements.

## Results

### Atomic diffractive optical elements concept
In Fig. 1a, we present the concept of our ADOEs. Two types of diffractive optical devices are implemented. First, in Fig. 1a, we illustrate a lamellar grating consisting of rectangular channels etched in silicon with rubidium vapor filling the channels and sealed shut with borosilicate glass. The period of the diffraction grating is ~50 μm, and the etch depth is ~150 μm. The portion of silicon in the period is ~30 μm, and the atoms are confined in channels of ~20 μm. Also illustrated in Fig. 1a is the diffraction pattern of light reflected from the grating surface. By scanning the frequency of the incident light, the diffraction efficiency is altered. For instance: two different detunings from the resonance frequency

changes the overall phase response of the atomic system eliciting distinct spatial diffraction patterns illustrated in red and green (shifted for clarity). Figure 1b illustrates an atomic Fresnel lens. Here, a series of circular channels with varying period to facilitate lensing action are etched in silicon. An atomic reservoir is connected to these circular gratings via an additional etched channel. The focal plane is also illustrated in this figure, with two different states corresponding to two different frequencies: an on-state, illustrated in red, and an off-state illustrated in green.

A photograph of a typical device is presented in Fig. 1c, where a few silicon-based Fresnel lens are connected to an atomic reservoir, which accommodates an alkali dispenser pill. A layout of such a device is presented in the Supplementary Fig. 1. The silicon frame (consisting of multiple optical elements) is anodically bonded to a borosilicate wafer, and a RbMbO$_x$/AlZr pill is heat-activated using a laser to release natural abundance rubidium post sealing (see the Methods section). Figure 1d presents a photograph of three typical devices after dicing.

### Atomic diffractive grating spectroscopy
We study the spectra of the atomic diffractive grating by first estimating the zeroth and first-order modes spectral response. By calculating[30] the rubidium bulk susceptibility and using it to introduce a phase-amplitude grating consisting of alternating columns of rubidium and silicon, we construct a theoretical diffraction response. Decomposing the bulk susceptibility into real and imaginary parts yields the amplitude and phase profiles, respectively, which are combined with the geometry of the device to fully define the phase-amplitude profile of the diffractive element. The frequency-dependent phase-amplitude grating imposed by the rubidium atoms is modeled by Fourier transforming (i.e., $F(a(\omega, r) \cdot e^{i\phi(\omega, r)})$, where F is the Fourier operator, and $a$ and $\phi$ are the spatially and frequency-dependent amplitude and phase response of the atoms, respectively) the phase-amplitude profile, from which we calculate the zero and first-order diffraction spectra, plotted in Fig. 2b. Figure 2a shows a reference absorption and dispersion spectrum of the D2 line (at a wavelength of 780 nm) of natural rubidium. Marked with dashed lines are the peak positions of the Doppler broadened absorption line centers of $^{85}$Rb and $^{87}$Rb; both spectra presented in Fig. 2b exhibit strong oscillations in spectral regimes far from absorption peaks (i.e., between the two sets of doppler broadened absorption peaks, and red or blue detuned from either peak). On the contrary, in the regime of absorptive features, a flat saturated response is predicted due to the high optical density of the atoms preventing a frequency-dependent response. The origin of such oscillations may be explained to be a result of the interferometric nature of the pure phase grating. In a symmetric square-phase grating, the zero-order diffraction intensity scales as the square cosine of the phase difference. Moreover, as is also evident from Fig. 2b, the first-order diffraction spectrum is in quadrature to that of the zero order, and exhibits an intensity that scales with the square sine of the phase. Indeed, such calculations demonstrate the ability to control the efficiency of diffraction by means of controlling the phase response of the atomic medium, at a given frequency detuning. The calculations presented here are based on a thin-element model and neglect the effect of the ~150 μm depth of the columns of atoms. Although, as will be shown in the next paragraph, such a model is highly effective at predicting qualitive features of the response of the grating, the thin-element model does not provide very accurate quantitative predictions. We improve this thin-element model by conducting Rigorous Coupled Wave Analysis (RCWA) calculations[31], using the MATLAB-based opensource code[32]. By doing so, we find that the qualitive response replicates the thin-element analysis (i.e., a cosine-like

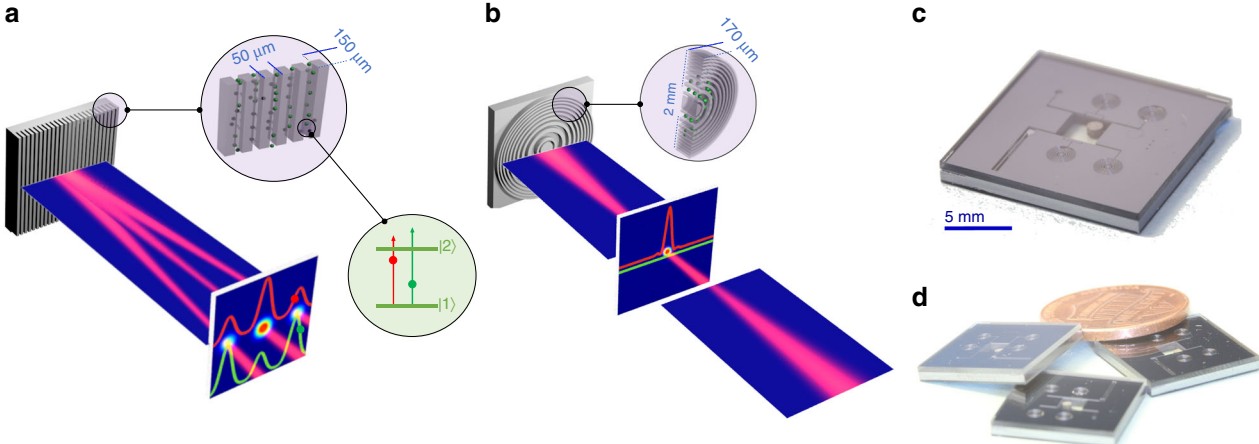

**Fig. 1** Atomic diffraction optical elements concept of operation. **a** Artistic rendition of a diffractive atomic grating whose diffraction pattern is controlled by the atomic state of atoms embedded within its channels. At the far-field plane, two different diffractive patterns are depicted corresponding to two different detuning from the optical transition (illustrated in the green circle). **b** Artistic rendition of an atomic switchable Fresnel lens. In the focal plane, two states of operation are shown, which correspond to two different detuning's from the atomic state: the red state corresponds to the lens in the on-state, and the green to the off-state. **c** A photograph of an ADOE consisting of a rubidium reservoir connected to a few manifestations of Fresnel lens. **d** A photograph of three typical diced devices compared to a penny

response for the zero order and a quadrature response for the other orders). Yet, the RCWA model predicts a quantitative variation in efficiency as a function of incident angle and polarization. We find that the incident angle has a severe impact on efficiency, which can vary from ~50 to 100%, within a window of ~1 degree change of incident angle. Moreover, according to these calculations, for a given small incident angle, the impact of polarization may be also significant. For instance, for an incident angle of 10° and a TM-polarization (i.e., rotating the polarization by 90°) the contrast is reduced to ~50%.

Figure 2c plots the experimental first- and zero-order spectra, measured at a temperature of ~180 °C, and spatially resolved using a photodetector and a pinhole. The data were normalized to the maximum reflection of the zero order as well as compensated for normally incident glass-air reflections. Generally, the fraction of reflected power to each of the orders was a few percent of the incident power. This efficiency agrees with the predicted RCWA calculations. Clearly, the same features predicted by the simple two-level Fraunhofer model, shown in Fig. 2b, describe the experimental data. Interestingly, the experimental data exhibit small peaks and dips (which depend on the order of diffraction) within the presumably optically thick absorption bands. The origin of such peaks most likely stems from a contribution of atoms near the front borosilicate window forming an atom–dielectric response. We note that although we analyze the zero- and first- order of diffraction, other nonzero orders of diffraction exhibited similar behavior. Here, we focus on the stronger odd orders of diffraction, although even orders may exist due to the deviation from a 50% duty cycle.

The diffraction efficiency can be modulated by up to 50% by altering the laser frequency. This change is a pure phase change (with a calculated absorption of less than 0.5%), as it is occurring at a detuning of 5 GHz to the red of the $^{85}$Rb transition. The reduced contrast in the fringes is almost certainly due to the nonuniformity of etch depth (see the Methods section), which gives rise to a loss of spatial coherence of the reflected light. Following, in Fig. 2d, we demonstrate the evolution of the first-order diffraction spectrum as function of atomic density. The atomic density is varied from a density of ~$3 \times 10^{12}$ cm$^{-3}$, (corresponding to a temperature of ~94 °C) to a density of ~$4 \times 10^{14}$ cm$^{-3}$ (corresponding to a temperature of ~184 °C). We have observed that higher densities and high laser intensities are

readily achievable and demonstrate effects such as energy pooling and radiative collisions[33], which may be useful for indirect frequency conversion. With the lower density, the first-order diffraction spectrum follows a typical absorption spectrum. By gradually increasing the atomic density, the spectra become more dispersive with spectral wings appearing between absorption bands finally evolving into oscillating patterns as discussed with respect to Fig. 2b and c. Interestingly, the frequency of such oscillations is directly related to the group index of the atomic medium in the so-called slow light regime between absorption peaks. Indeed, such a relation between group index and these features has been studied in the context of slow light-enhanced interferometry[34]. As we increase the density of atoms, we expect self-broadening and wall collisional broadening to become more pronounced, and thus affect the homogeneous linewidth. For instance, at a density of ~$4 \times 10^{14}$ cm$^{-3}$, one should expect an order of magnitude increase of the natural linewidth. This effect somewhat reduces the rate that the group index increases with density, but has a relatively small impact on the spectra.

**Atomic Fresnel lens**. The second element we demonstrate is the atomic Fresnel lens. As previously discussed, the lens constitutes annular concentric silicon rings with rubidium filling the space between the rings. The lens, designed to have a focal point of 70 mm, has a dimeter of 2 mm and the distance between rings ranges from 40 μm to 120 μm. Characterization of a bare lens (i.e., prior to activating Rb atoms) is presented in the Supplementary Fig. 1, as well as a typical layout of the lens. A photograph of our device is presented in Fig. 3a. Here, a few Fresnel lenses which differ in size and topology are implemented in the same device. A section of the photograph is shown in the same figure where the rubidium dispenser connecting channels and two Fresnel lenses are visible.

We illuminate the lens with 780 nm light resonant with the D2 resonances in rubidium and scan the laser about the absorption bands. We position a CCD camera at the measured focal plane of the Fresnel lens in absence of atoms. Figure 3a illustrates the optical power incident on a single pixel as a function of frequency detuning relative to $^{85}$Rb strongest resonance dip. The single pixel is chosen to coincide with the maximal power incident on the CCD. Figure 3b illustrates a series of cross-sections of the optical

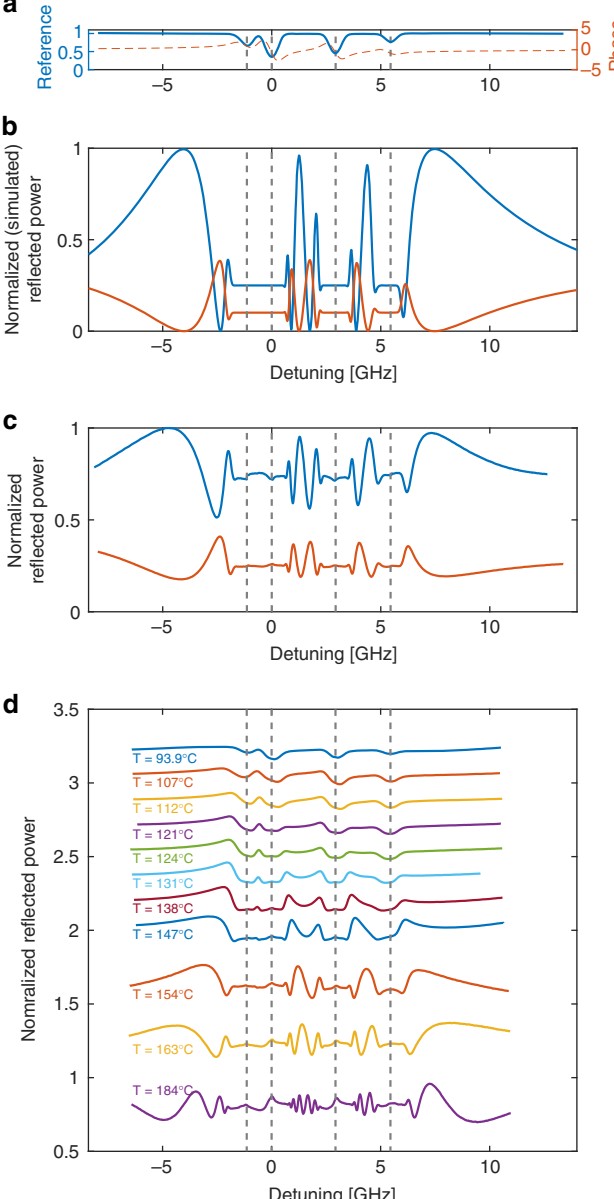

**Fig. 2** Atomic diffraction grating spectroscopy. **a** Reference D2 rubidium absorption and calculated phase spectrum. The 0 GHz detuning is in reference to the $^{85}$Rb F = 3 state, and the phase plot is in units of $2\pi$. **b**, **c** Spectra of zero-order (blue) and first-order (orange) atomic diffraction grating, measured by combining a photodetector and a pinhole in the far-field (**b**) calculated (**c**) measured spectra (**d**) evolution of the measured first-order spectra as a function of atomic density, demonstrating the evolving of the spectra from being absorptive to dispersive

power profile incident on the CCD as the distance between the atomic lens and the CCD is changed. Figure 3c and d illustrates two different frequency detunings, −3.4 GHz and −2.4 GHz, respectively, which strongly modify the properties of the lens and effectively switch between focusing and not focusing the beam at the designed focal plane. A more elaborate evaluation of the effect at other detunings is presented in the Supplementary Fig. 2, as well as in Supplementary Movie 1 showing this evolution of focal plane intensity with detuning. The contrast of the optical power at these two frequencies exceeds 13 dB, and is equally illustrated in the large optical power swings seen in Fig. 3a. As can be expected, the grating behavior at high optical densities when the

laser is on the atomic resonances is strongly absorbed and does not contribute to lensing action. Conversely, far off-resonance, the steep dispersion of the atoms strongly modulates the phase difference between the different Fresnel zones which allows the lens to be switched between focusing and defocusing actions. The improved contrast in this device, as compared with the results in Fig. 2 can be attributed to a fabrication process applied to the Fresnel lens ADOEs which improves the uniformity of the etch depth (see description in the Methods section).

**Nonlinear and magnetic-optical control of ADOEs**. Alkali atomic vapors are highly versatile material systems, with unique properties including the ability to control the atomic medium all-optically and with external magnetic fields. As an example, uti-lizing atomic vapor in confined geometries, has shown the ability to control light all-optically with minute levels of optical power[16,35]. Another example includes the demonstration of a highly compact optical isolator, exploiting the exceptionally high Verdet constant of Rb[17]. To explore the nonlinear properties of ADOEs, we study the spectra of the atomic grating when chan-ging the applied incident laser power. By doing so, we control the degree of optical pumping and saturation of our structured atoms and in turn change the dispersive properties of the atomic medium. In Fig. 4a, we present such spectra where we record a portion of the ADOE reflection spectrum (recorded at a slight angle of a few degrees) corresponding to the oscillating features present between the two absorption lines of $^{85}$Rb. Here, two different spectra are shown in blue and red, corresponding to 10 mW and 10 µW of power, respectively. As is evident from this figure, the period of the fringes is increased as a direct result from nonlinear modification of the atomic group index. We claim this increase in the period to be a result of the optically induced reduction in the amplitude of the atomic refractive index profile. However, one could speculate the origin of this effect to be a result of optical heating, which would increase the density of atoms, or thermo-optically alter the grating transfer function. Yet, an increase of atomic density would induce an opposite effect. If one would attribute this reduction to a spectrally uniform change of density of atoms our result corresponds to a reduction of the temperature of the atoms of ~3 °C corresponding to a ~13% change in the atomic density. As to the thermal-optical effect, even significant (~Δ100 °C) optical heating the silicon frame which would increase the complex refractive index of the silicon frame is expected to induce a small change in the phase and contrast of the fringes. Yet, such a thermo-optical effect is not expected to change the period of the fringes, which is what we observe experimentally. This expectation is also confirmed by RCWA calculations. Moreover, we have witnessed a significant reduction in the Doppler-free peak shown in Fig. 2d confirming that the witnessed effect is indeed originating from the nonlinear light–vapor interaction. In terms of a frequency shift, this change may manifest in a change as large as 70 MHz (see the fringes at a detuning of ~1 GHz) or is nulled at the points where the two curves intersect. We stress that this change is predominantly a phase change, which has a negligible absorption component as is evidenced by the intact overall fringe envelope. The wide range of optical fringes can be used as a tunable offset frequency lock and is currently being explored.

The interaction of an atomic medium with magnetic fields is important in optical magnetometry, as well as an efficient means to control the state of polarization of light. Here, we apply a moderate (~300 G, directed ~45° to the normal of the device plane) static magnetic field using a bar magnet to the ADOE grating structure. In Fig. 4b, we present a portion of the reflected spectra of the ADOE grating in the presence (blue lines) and

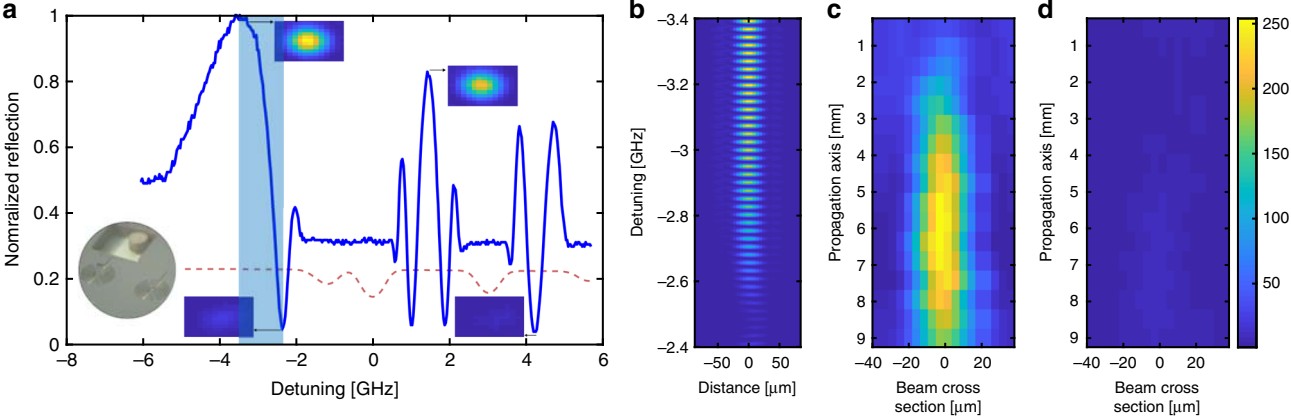

**Fig. 3** Atomic Fresnel lens spectroscopy and characterization. **a** Fresnel lens spectrum at the focal point, obtained by post processing a set of recorded CCD images while simultaneously scanning the lasers frequency. Inset images are examples of such images at different frequency detunings. An additional inset shows a zoomed photograph of the actual device. **b** A series of CCD images recorded at the focal plane corresponding to the maximal position with detuning of −3.4 GHz to the minimal point with detuning of −2.4 GHz. **c, d** The spatial dependence of the focal point cross-section as function of the propagation axis plotted for the two extreme (**c**) on and (**d**) off states. Scale bar refers to figures **b–d** and is in units of normalized reflected power

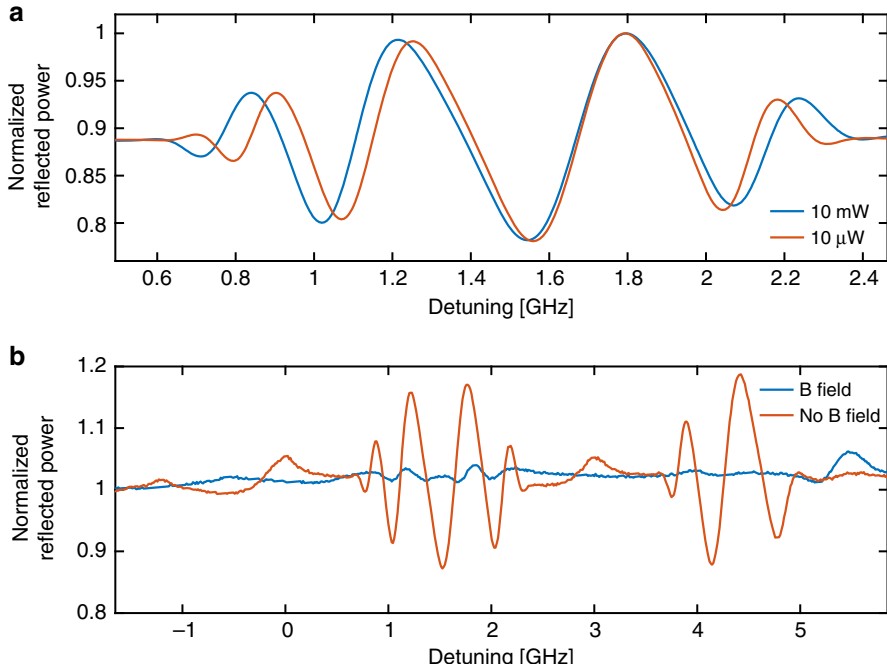

**Fig. 4** ADOE response as function of laser power and external magnetic field. **a** Normalized first-order reflected power as a function of frequency detuning of an atomic diffractive grating, with incident power of 10 mW (blue line) and 10 μW (red line). A portion of the spectra corresponding to spectroscopic data between the two absorption lines of $^{85}$Rb is shown here. **b** Normalized first-order reflected power as a function of frequency detuning of an atomic diffractive grating for the case where a ~300 gauss magnetic field is applied to the grating (blue lines) and in the absence of such field (red line)

absence of the magnetic field (red lines). Clearly, the magnetic field almost totally diminishes the oscillating response of the atomic grating. We attribute the dominant effect of the diminished response to the creation of a phase gradient across the grating induced by the magnetic field gradient. Fundamentally, the magnetic field splits the atomic transition energies via the Zeeman effect. As a consequence, the phase response for a given frequency changes as function of magnetic field strength. Thus, the magnetic field encodes a varying phase response for each atomic channel that constitutes the grating, which strongly affects the phase coherence of the ADOE. This effect may be used for sensing of a magnetic field gradient across the different channels constituting the AODE grating through constructive interference much like in a phased array. Since we operate this

experiment with a small incident angle, a rotation of the polarization (originating from magnetically induced circular birefringence) may also contribute to a reduction of the overall ADOE grating contrast, as we have calculated with the RCWA model mentioned above.

## Discussion

To summarize, we have presented the concept of chip-scale, quantum diffractive optical elements. Consisting of traditional microscale diffraction optics embedded with rubidium vapor, the platform provides a unique system capable of mapping atomic properties onto the outputs of diffractive optical elements. We have demonstrated two distinct diffractive elements. The lamellar

atomic grating demonstrates efficient control of the spatial distribution of light in the far field by controlling the phase of light interacting with the atoms within the grating. Moreover, we studied the atomic diffraction grating as functions of diffraction order, atomic density, and frequency and found them to agree well with a simple model. By inheriting the same design features of the atomic diffraction grating, we demonstrate a quantum Fresnel lens. By characterizing the lenses' spatial and frequency-dependent properties, we demonstrate efficient switching of the focusing properties of the lens with greater than 95% contrast by changing the frequency of the laser by only 1 GHz. Finally, we have demonstrated the ability to control ADOEs exploiting either the optical nonlinearity of the atomic medium or using static magnetic fields, and found that we can modify the behavior of the ADOE in both cases, opening a clear path for all-optically and magnetically controllable ADOEs.

The incorporation of atomic vapors into DOEs offers unique properties to the overall ADOE, when compared with other tunable DOE material systems[36,37]. By encoding the atomic energy-level structure into the optical transfer function, one gains access to a fundamental material system offering atomically derived, steep phase and amplitude gradients. Exploiting such properties enables one to explore intriguing applications spanning from highly compact atomically derived frequency references and accurate optical spectrometers to advanced dispersive imaging systems. For instance, in view of Fig. 2, one can exploit the off-resonance dispersive response of the ADOE to implement an offset-lock similar to dichroic atomic vapor laser lock (DAVLL). Such a technique may find applications in many different quantum sensing experiments. In this work, we have exploited the frequency-dependent refractive index change of the atomic medium, and explored its nonlinear optical properties and susceptibility to magnetic fields. Yet, other degrees of freedom such as chirality, entanglement, quantum coherence, and two-tone all-optical control may further unravel exciting applications and physics.

Providing the design tools for chip-scale atomic diffractive optical elements draws a path for a variety of compact, thin quantum-optical elements, and quantum metasurfaces which may have an important impact on a myriad of fields within metrology and quantum technologies.

## Methods

**Device fabrication and filling**. Two different processes were explored for wafer-level device fabrication. The first, starting from a 2 -mm-thick silicon 4" wafer, used deep reactive ion etching (DRIE) to blind etch the grating pattern into bulk silicon. Typical etch depths of 150 μm were used in the devices reported here with aspect ratios of ~10. Following, a 1.5 -mm-deep reservoir area was etched connecting to previously etched channels. This first process had a nonuniform etch depth due the lack of an etch stop. With the second process, the same etching procedures were implemented exchanging the pure silicon wafer with a silicon on silicon-dioxide wafer. These wafers had three layers consisting of a 170 -μm-thick silicon upper layer separated from an 830 -μm silicon handle by a 200 -nm silicon-dioxide layer. Using the silicon dioxide as an etch-stop layer, etch depth uniformity is significantly improved. Following etching, a wet buffered oxide etch removes the exposed silicon dioxide. Finally, a second DRIE process is used to define the reservoir.

To introduce rubidium and seal the device, commercial rubidium dispenser pills consisting of a RbMbO$_x$/AlZr reducing agent[29] are added to the etched reservoirs. Following, we bond a borosilicate wafer to the top (and bottom, for the case of the second type of process) of the silicon structure using a commercial anodic bonder (AML Wafer Bonder). We activate the dispenser pill to release rubidium into the cell by using ~1 W of 980 -nm laser power weakly focused on the dispenser pill. Finally, the wafer is diced to achieve cm-scale lateral dimension devices.

**Optical setup**. To characterize the ADOEs, we use a 780 nm DBR (Photodigm) laser operating at the D2 line of rubidium. Devices were heated using resistive heaters which generate small thermal gradients across the chip to avoid condensation of rubidium in the channel areas. Clogging of the channels was not observed in devices near the hot or cold side of the thermal gradient. A collimated laser beam of ~2 -mm diameter was directed to illuminate the specific device under test and the reflected diffracted beam is detected at normal incidence using a nonpolarizing beam splitting cube directing light through a pinhole and onto a CCD camera or a photodiode. While scanning the laser across the D2 manifold, a reference cell spectrum is recorded.

## Data availability
The data and code that support the findings of this study are available from the corresponding author on reasonable request.

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

## Acknowledgements

The authors acknowledge Peter Lowell, Jim Nibarger, and James Beall for discussions and assistance in fabricating the devices, and Azure Hansen and James McGilligan for comments on the paper. This work is a contribution of the US government, and is not subject to copyright in the United States of America. Note: Any mention of commercial products within this letter is for information only and does not represent an endorsement from NIST.

## Author contributions

L.S. conceived the concept and analyzed the data; L.S. and D.G.B. performed the experiments and wrote the paper; L.S., D.G.B., V.N.M. and S.A.S. contributed to fabrication of devices; J.E.K. supervised the project, helped design the experiments, and helped write the paper.

## Additional information

**Competing interests:** The authors declare no competing interests.

