## [Peer Review File · Nature Communications]

Reviewers' comments:

Reviewer #1 (Remarks to the Author):

In the revised version of the paper, the author have responded adequacy to all the issued I have raised in the review. As such, I find this paper suitable for publication in Nature Comm. Clearly, it will be an important addition to the emerging field of micro/nanoscale light vapor interactions.

Reviewer #2 (Remarks to the Author):

Dear Editor,

The authors have adequately addressed my previous comments. They have also significantly enhanced their manuscript by demonstrating active tuning of their devices by laser irradiation and external magnetic field. Based on that I would like to recommend the publication of this manuscript in Nature Communications after the authors address the following comments related to the newly introduced manuscript parts:

- 1) When increasing the laser power up to 10 mW, how the authors can be sure that the observed spectral shift effect is not due to the heating of the silicon grating itself? 780 nm is well absorbed in silicon and 10mW is a high-enough power to heat the sample, while silicon is well known for its thermo-optical effects.
- 2) The effect of magnetic field on the observed signals is not well explained. At least it cannot be easily understood by a non-specialist in the field. The authors should clearly explain, how magnetic field interacts with the atomic vapor and how does this translate to the observed spectrum.
- 3) The numbers in Figure Supplementary 1 are not clearly visible and should be increased in size.

Point by point response to both reviewer's comments:

Reviewers' comments:

Reviewer #1 (Remarks to the Author):

In the revised version of the paper, the author have responded adequacy to all the issued I have raised in the review. As such, I find this paper suitable for publication in Nature Comm. Clearly, it will be an important addition to the emerging field of micro/nanoscale light vapor interactions.

Thank you very much for your valuable comments and finding our paper suitable for publication.

Reviewer #2 (Remarks to the Author):

Dear Editor,

The authors have adequately addressed my previous comments. They have also significantly enhanced their manuscript by demonstrating active tuning of their devices by laser irradiation and external magnetic field. Based on that I would like to recommend the publication of this manuscript in Nature Communications after the authors address the following comments related to the newly introduced manuscript parts:

1) When increasing the laser power up to 10 mW, how the authors can be sure that the observed spectral shift effect is not due to the heating of the silicon grating itself? 780 nm is well absorbed in silicon and 10mW is a high-enough power to heat the sample, while silicon is well known for its thermo-optical effects.

Thank you very much for your very valuable comment. The effect of heating the silicon grating may have two effects: The first, to change the complex refractive index of the silicon, and change the transfer function of the grating. The second, would be the increase of the density of atoms. For the first effect, we expect such increase in refractive index to induce a change in the contrast of the fringes as well as their phase. Yet, such thermo-optical effect is not expected to change the period of the fringes (as we witness in experiment). This expectation is also confirmed by RCWA calculations. The increase of the density of the atoms, would in fact increase the group index, and increase the number of fringes we see. In experiment, we observe a reduction of the group index and number of fringes, which we calculate to *reduce* our effective atomic temperature by 3°C. Moreover, we have witnessed a significant reduction in the Doppler-free peak shown in Fig. 2d confirming that the witnessed effect is indeed originating from the non-linear light-vapor interaction. We have now added a paragraph (page 5) which addresses this comment.

2) The effect of magnetic field on the observed signals is not well explained. At least it cannot be easily understood by a non-specialist in the field. The authors should clearly explain, how magnetic field interacts with the atomic vapor and how does this translate to the observed spectrum.

Thank you for this important comment. We have now added a paragraph to address this comment on page 5.

3) The numbers in Figure Supplementary 1 are not clearly visible and should be increased in size.

Thank you, we have fixed this.

REVIEWERS' COMMENTS:

Reviewer #2 (Remarks to the Author):

The authors have adequately addressed all my comments and the manuscript can now be accepted for the publication.

Point by point response to both reviewer's comments:

Reviewers' comments:

Reviewer #2 (Remarks to the Author):

The authors have adequately addressed all my comments and the manuscript can now be accepted for the publication.

Thank you very much for your valuable comments and finding our paper suitable for publication.